# Effects of Selective Substitution of Cysteine Residues on the Conformational Properties of Chlorotoxin Explored by Molecular Dynamics Simulations

**DOI:** 10.3390/ijms20061261

**Published:** 2019-03-13

**Authors:** Andrew J. Gregory, Leah Voit-Ostricki, Sándor Lovas, Charles R. Watts

**Affiliations:** 1Department of Neurosurgery, Mayo Clinic Health System-Franciscan Healthcare in La Crosse, La Crosse, WI 54601, USA; ajgregory2@wisc.edu (A.J.G.); voit-ost.leah@uwlax.edu (L.V.-O.); 2Department of Biomedical Sciences, Creighton University, Omaha, NE 68178, USA; SandorLovas@creighton.edu; 3Department of Neurologic Surgery, Mayo Clinic, Rochester, MN 55905, USA

**Keywords:** αβ motif, Abu, chlorotoxin, Cys, disulfide bond, insectotoxin, isosteric substitution, l-α-aminobutyric acid, molecular dynamics, Ser

## Abstract

Chlorotoxin (CTX) is a 36–amino acid peptide with eight Cys residues that forms four disulfide bonds. It has high affinity for the glioma-specific chloride channel and matrix metalloprotease-2. Structural and binding properties of CTX analogs with various Cys residue substitutions with l-α-aminobutyric acid (Abu) have been previously reported. Using 4.2 µs molecular dynamics, we compared the conformational and essential space sampling of CTX and analogs with selective substitution of the Cys residues and associated disulfide bonds with either Abu or Ser. The native and substituted peptides maintained a high degree of α-helix propensity from residues 8 through 21, with the exception of substitution of the Cys^5^–Cys^28^ residues with Ser and the Cys^16^–Cys^33^ residues with Abu. In agreement with previous circular dichroism spectropolarimetry results, the C-terminal β-sheet content varied less from residues 25 through 29 and 32 through 36 and was well conserved in most analogs. The Cys^16^–Cys^33^ and Cys^20^–Cys^35^ disulfide-bonded residues appear to be required to maintain the αβ motif of CTX. Selective substitution with the hydrophilic Ser, may mitigate the destabilizing effect of Cys^16^–Cys^33^ substitution through the formation of an inter residue H-bond from Ser^16^:OγH to Ser^33^:OγH bridged by a water molecule. All peptides shared considerable sampled conformational space, which explains the retained receptor binding of the non-native analogs.

## 1. Introduction

Chlorotoxin (CTX) is a peptide toxin in the venom of the deathstalker scorpion (*Leiurus quinquestriatus*) [1,2]. The peptide binds with high affinity to chloride channels, causing paralysis in invertebrates, but it has minimal to no effect on vertebrates or mammals [2]. Because of its high affinity and selectivity, CTX was originally used as a tool to characterize the function of chloride channels in electrophysiology experiments. Pharmacologic interest in the peptide increased after CTX was shown to bind with high affinity to glioma-specific chloride channels on the surface of World Health Organization Grade IV intrinsic brain tumors, as well as other tumors of neuroectodermal embryologic origin [3,4,5]. CTX has been considered a potential lead for the development of novel therapeutic agents, imaging adjuncts, and intraoperative optical imaging “tumor dyes/paints” [6,7,8,9,10].

CTX consists of 36 amino acids with eight Cys residues at positions 2, 5, 16, 19, 20, 28, 33, and 35 (I–VIII for homology modeling) that form four disulfide bonds between residues 2–19 (I–IV), 5–28 (II–VI), 16–33 (III–VII), and 20–35 (V–VIII) (Figure 1); Roman numerals in parentheses refer to the homology numbering [11,12,13,14,15,16,17,18,19,20]. The resulting secondary and tertiary structure are known as an αβ (βαββ) motif, a folding scaffold common to insectotoxins, insect defensins, plant γ-thionins, and inhibitory cystine knot peptides [11,12,13,14,15,16,17,18,19,20]. Multi sequence alignment of 20 scorpion toxin–derived peptides show that they share 49% to 88% sequence similarity [11,12,13,14,15]. Overlays of the three-dimensional structures of several of these peptides results in root-mean-square deviation (RMSD) less than 0.1 nm, which indicates the stability of the αβ motif [21]. The observed differences in channel blocking and molecular selectivity of the toxins are due to subtle differences in residue charge and sidechain conformation that affect surface electrostatic charge distribution and complementarity [21].

The locations of the disulfide bonds within this family of peptides are important for maintaining secondary and tertiary structural features and protease resistance [16,22,23,24,25,26]. Despite the evolutionarily conserved disulfide bonds in this family of insectotoxins, the ability to selectively remove some of these bonds is well documented [24,25,26,27,28]. The II–VI bond of charybdotoxin and leiurotoxin I can be selectively removed by substitution of Cys residues with l-α-aminobutyric acid (Abu) without significant effects on oxidative folding [24,25,26,27,28]. Ojeda et al. demonstrated that selective substitution of Cys residues involving either the I–IV or II–VI disulfide bonds of CTX with Abu residues had little effect on peptide conformation, whereas the III–VII and V–VIII disulfide bonds were critical to the process of oxidative folding and obtaining nativelike peptides [29]. Complete substitution of all disulfide-bonded Cys residues resulted in a peptide that maintained its biological activity with total loss of native secondary and tertiary structure and significantly increased susceptibility to serum proteases [29].

The tertiary structures of CTX in water (Protein Data Bank ID: 1CHL) has been determined using high-field ^1^H nuclear magnetic resonance spectroscopy (NMR) (Figure 2) [30]. The presence of the N-terminal β-sheet from Cys^2^ to Cys^5^ is variable and dependent on the algorithm used to determine the structures [31,32]. In the current study, we used molecular dynamics (MD) simulations of CTX and its Abu- and Ser-substituted analogs to investigate the role of the disulfide bonds in stabilizing the αβ motif (Table 1). Emphasis was placed on investigating the role of hydrophobic (Abu) versus hydrophilic (Ser) isosteric substitutions, the distribution of the hydration shell around each respective residue substitution, and subsequent changes in secondary and tertiary structures.

## 2. Results

### 2.1. General Properties

The RMSD_CTX_, RMSD_AVG_, and fractions of sampled DSSP secondary structure (α-helix, β-sheet, and β-bend/turn), for CTX and Abu- and Ser-substituted analogs are given in Table 2. The values of RMSD_CTX_ and RMSD_AVG_ were similar between CTX and all analogs (Table 1), with the exception of CTX5(A) and CTX5(S), which had lower RMSD_AVG_ values.

The fraction of α-helix is retained across all analogs with the lowest fraction corresponding to CTX2(A) and the highest to CTX1(A) (Table 1). Likewise, the fraction of β-sheet is retained in the analogs where a single disulfide bond is substituted but markedly decreased in the CTX5(A) and CTX5(S) analogs. The greatest degree of variability in global secondary structure occurs for β-turn/bend. Although the fraction of α-helix and β-turn/bend for CTX5(A) was 0.28 and 0.21, respectively, the fraction of random coil was 0.48, which is consistent with previously published CD spectra [29].

### 2.2. DSSP Secondary Structure

The residue fractions of sampled DSSP secondary structure for CTX and each of the Abu- and Ser-substituted analogs are shown in Figure 3. The secondary structure for CTX was consistent with its ^1^H NMR solution conformation, having 3 prominent β-turns/bends in the N-terminal domain from residues 6 through 9 and 12 through 13, an α-helix from residues 15 through 20, and 2 antiparallel β-sheets from residues 27 through 29 and 32 through 34, with intervening β-turns/bends from residues 21 through 24 and 30 through 31. The N-terminal predicted β-sheet from residues 2 through 5 was absent. For all single disulfide bond-substituted analogs, the β2, β3 antiparallel region is well preserved. The α-helix, although present, is more variable in its location, length, and populated fraction. The CTX5(A) and CTX5(S) analogs demonstrate disruption of the native CTX αβ motif with elongation of the N-terminal α-helix, loss of the β2, β3 antiparallel region and increases in β-turn/bend sampling throughout.

### 2.3. Structural Flexibility

The Cα-trace RMSFs comparing each of the analogs with the average conformation of CTX are shown in Figure 3. There were only minor differences in the Cα-trace RMSF between CTX and that of its average conformation. The most flexible regions of CTX correspond to the β-turn/bend regions from residues 8 through 9, 12 through 13, 21 through 24, and 30 through 31. All single disulfide bond Abu-substituted analogs demonstrated increased residue flexibility compared to native CTX, while the Ser-substituted analogs did so to a much lesser degree. The decrease in Cα-trace RMSF was most marked for the CTX1(S) and CTX4(S) analogs corresponding to a lengthening of the α-helix region. As expected, the CTX5(A) and CTX5(S) analogs show the greatest degree of Cα-trace RMSF deviation from native CTX.

### 2.4. Interactions

The backbone–backbone (BB–BB) and sidechain–sidechain (SC–SC) contact probability maps are shown in Figure 4. For CTX, most of the high-probability contacts are SC–SC, with the expected long-distance contacts between the 2–9; 5–28; 16–33; or 20–35 disulfide-bonded residues. The SC–SC contacts between the N-terminal residues with the α-helix from residues 15 through 20 and the 2 C-terminal β-sheets from residues 27 through 29 and 32 through 34 are also present. The only long-distance BB–BB contacts are between the 2 antiparallel β-sheets and interactions between residues 4 and 5 with 31 through 33. In CTX1(A) both SC–SC and BB–BB contacts are increased. The removal of the Cys^2^-Cys^19^ disulfide bond facilitated increased interactions between the N-terminal region, the α-helix, and the C-terminal antiparallel β-sheets. There were also increased BB–BB interactions between the α-helix from residues 12 through 17 and the proximal β-sheet from residues 24 through 28 and between the N-terminal 8 residues and residues 28 through 32. The SC–SC and BB–BB contact probabilities in CTX1(S) and CTX2(A) were similar to those in CTX. Contacts in CTX2(S) were similar to those in CTX1(A) except that the interaction between the N-terminal residues and α-helix with the proximal β-sheet were much less pronounced and an interaction was present between the N-terminal residues 1 through 8 and the α-helix from residues 13 through 19. Contact probability maps for CTX3(A) and CTX3(S) were similar and shared similarity with CTX2(S). Likewise, CTX4(A) and CTX4(S) were similar to CTX. CTX5(A) had sparse SC–SC interactions and low-frequency BB–BB interactions between multiple residues. CTX5(S) was the exact opposite, with few, if any, BB–BB interactions and sparsely populated SC–SC interactions.

To determine the degree conformational stability conferred by each disulfide bond on CTX and its Abu- and Ser-substituted analogs, the CβHβ^1,2^–CβHβ^1,2^ center-of-mass distances between residue pairs and probabilities of contact ≤ 0.26 nm are shown in Table 3. The CβHβ^1,2^–CβHβ^1,2^ center-of-mass distance was used because with ^1^H NMR, the method of structural determination used for CTX, S–S interatomic distances cannot be assigned in spectra. The distance geometry algorithm calculation of peptide conformations is therefore dependent on the presence of bonded CysHβ^1,2^–CysHβ^1,2^ nuclear Overhauser effect spectral peaks. Removal of a disulfide bond in most of the substituted peptides resulted in significant increases in the CβHβ^1,2^–CβHβ^1,2^ center-of-mass distances and decreases in the probabilities of contact, with the exception of CTX3(S) and CTX4(A).

The probabilities of finding water molecules within 0.5 nm of the terminal γ-sidechain atoms (Sγ, CγHγ, and OγHγ) of Cys, Abu, and Ser respectively, and the residue relative solvent-accessible surface area (rSASA) of the residues are shown in Table 4. The terminal Sγ of CTX remained relatively solvent shielded, regardless of its position within the native peptide with the exception of the Cys^2^, Cys^19^ and Cys^35^ residues. The Cys^2^ and Cys^35^ residues are located at the more solvent-accessible N- and C-termini respectively while the Cys^19^ residue is on the solvent exposed surface of the N-terminal α-helix, Figure 2. Substitution of disulfide-bonded Cys residues with either a hydrophobic Abu or hydrophilic Ser at the 2,19 (I–IV), 5,28 (II–VI), and 20,35 (V–VIII) positions tended to increase the probability of adjacent water molecules at the involved residues and increase the rSASA. This local relationship does not however hold for the 16,33 (III–VII) disulfide bond. For the hydrophobic Abu-substituted CTX3(A) there is the expected increase in probability of adjacent water and rSASA for the substituted residues but also a slight increase for Cys^2^ and more interestingly, decreases for Cys^19^. For the hydrophilic Ser-substituted CTX3(S), the substituted residues demonstrate an increase in the probability of an adjacent water molecules and a significant decrease in rSASA indicating that the interacting water molecules may be sequestered from the bulk solvent. The CTX5(A), and CTX5(S) peptides showed significant increases in the probability of adjacent water molecules and increases in rSASA through the substituted residues.

### 2.5. Conformational Analysis

The lowest energy conformations were determined by projecting each trajectory onto the first 2 dihedral principal (dPC) components, as show in Figure 5. The lowest energy conformation of CTX was consistent with the ^1^H NMR structure as follows: α-helix from residues 15–20, 2 antiparallel β-sheets from residues 27–29 and 32–34, and 2 intervening turns from residues 22–25 and 30–31. There were 4 salt bridges: Arg^14^ and Asp^17^, Lys^15^ and Asp^18^, Arg^25^ and the C-terminus, and Lys^27^ and the C-terminus.

The lowest energy conformations of CTX1(A), CTX1(S), CTX2(A), and CTX3(S) maintain secondary and tertiary structures that are consistent with the αβ motif of native CTX. This is despite the finding that the sidechains of substituted residues remain separated and do not interact (Table 3). The CTX3(S) peptide is unique in that the two Ser residues are in close proximity (Table 3) and most likely interact by hydrogen bonding to each other or through an associated water molecule that is sequestered from the solvent, given the significant increase in ρ_g(r)_ and low rSASA (Table 4). The lowest energy conformations of CTX2(S), CTX3(A), CTX4(A), CTX4(S), CTX5(A), and CTX5(S) deviate significantly from the αβ motif and CTX, with changes in both their secondary and tertiary structure.

### 2.6. Essential Subspace Analysis

Figure 6 shows a comparison of sampled essential subspace and its normalized values. Despite the substantial changes that occurred in secondary and tertiary conformations with some of the selective and global Cys substitutions, particularly CTX2(S), CTX3(A), CTX4(S), CTX5(A), and CTX5(S), none of the nRMSIP values were 0.65 or less. This indicates that a significant degree of sampled conformational space is shared across all substituted peptides. The results can be divided into 4 classes for comparison: 0.65 ≤ nRMSIP < 0.75, 0.75 ≤ nRMSIP < 0.85, and 0.85 ≤ nRMSIP < 0.95, and 0.95 ≤ nRMSIP. Peptide pairwise comparisons sharing the greatest degree of subspace conformational sampling were CTX to CTX1(S), CTX to CTX3(A), CTX to CTX4(A), CTX1(S) to CTX2(A), CTX1(S) to CTX3(A), and CTX3(A) to CTX4(A).

## 3. Discussion

We performed µs-scale MD simulations to compare the conformational and essential space sampling of CTX and its analogs with selective substitution of the Cys residues in disulfide bonds with either Abu or Ser. The choice of Abu for substituting Cys residues has been driven by its isosteric and hydrophobic nature, with the goal of preserving the hydrophobic pocket in the peptide [27,33]. Because of the small size and highly charged nature of these peptides, the surrounding environment of the disulfide bonds may not be as hydrophobic as previously considered. In the current MD simulations, we were able to give detailed descriptions of the local hydrophobic/hydrophilic environment of each disulfide bond. Examination of the probability of water contact and rSASA data (Table 4) demonstrates that some of the Cys residues are relatively solvent shielded (5, 16, 20, 28, and 33) while others are more exposed (2, 19, and 35). For Cys^2^ and Cys^33^ the degree of solvent exposure and hydration is secondary to their location at the N- and C-termini respectively while Cys^19^ is more solvent exposed due to its location on the outside surface of the N-terminal α-helix. For Cys^5^, there may some degree of local solvent shielding contribution secondary to the adjacent hydrophobic Pro^4^ and aromatic Phe^5^ residues. The remaining Cys residues (16 and 20) however are adjacent to charged residues such as Lys, Asp, or Arg, with the exception of Cys^28^ which as adjacent to the charged Lys^27^ and aromatic Tyr^29^, and Cys^33^ which is adjacent to a hydrophilic Gln^32^ and hydrophobic Leu^34^.

Replacement the disulfide-bonded Cys^2^and Cys^19^ residues with either Abu or Ser has only minor effects on CTX secondary and tertiary conformations. Removal of this bond is well tolerated because the remaining Cys^5^–Cys^28^ disulfide bond maintains the N-terminus in close proximity with the antiparallel β-sheet. This interaction is also stabilized with an H-bond between the amine of Cys^5^ and the carbonyl of Pro^31^. Subtle differences are present between the Abu- and Ser- substituted analogs; CTX1(A) and CTX1(S). Abu substitution facilitates wide separation of the N-terminus from the α-helix and β-sheet. Ser-substitution allows for close approximation of the N-terminus to the distal strand of the β-sheet, forming a 3-strand antiparallel β-sheet. The CTX1(S) structure is stabilized by H-bonds between the amine of Cys^5^ and the carbonyl of Pro^31^, the amine of Cys^33^ and the carbonyl of Met^3^, the amine of Met^3^ and the carbonyl of Cys^33^, and NεH of Gln^32^ and Oγ of Ser^3^. The sidechain of the Ser^19^ residue on the α-helix is positioned so that it can interact with the surrounding solvent. Substitution of the Cys^2^-Cys^19^ disulfide bond with Ser residues most likely removes some degree of tertiary strain within the peptide. Furthermore, it allows more H-bonding interactions between the N-terminus and the antiparallel β-sheet. This is consistent with the disulfide bond between Cys^2^–Cys^19^ being more solvent exposed and thus favoring potential hydrophilic substitutions.

Substitution of the disulfide-bonded Cys^5^ and Cys^28^ residues allows the α-helix region to be lengthened from residues 9 through 20. The preserved disulfide bond between Cys^2^ and Cys^19^, however, maintains the N-terminal region in close proximity with the C-terminal β-sheets. Due to elongation of the α-helix, the N-terminal region is not oriented in a way that would allow H-bonding to the distal strand of the β-sheet. CTX2(A) places Abu^5^ and Abu^28^ residues within a hydrophobic cleft between the α-helix and the proximal strand of the antiparallel β-sheets interacting with the sidechain of Met^12^. This preserves the hydrophobic region present in CTX but in a slightly less compact conformation. The conformational importance of this hydrophobic region is demonstrated by Ser-substitution, CTX2(S), which results in significant disruption of the α-helix with a large loop extending from Ser^5^ through Lys^15^ and a shift in the location of the antiparallel β-sheets from residues 25 through 29 and 32 through 36. This results in a conformation that exposes the involved residues to solvent and disrupts the native hydrophobic region.

Despite the high degree of solvent shielding at the disulfide bond between Cys^16^ and Cys^33^, substitution of these residues with the hydrophobic Abu is not tolerated. This is supported by the detected non-native isomers in oxidative folding experiments [28,29]. There is significant structural disruption in the CTX3(A) peptide. The lack of a disulfide bond between Cys^16^and Cys^33^ disrupts the association between the α-helical region and the terminal strand of the antiparallel β-sheets. This is particularly important because the sidechains of the antiparallel β-sheet residues are oriented away from the α-helix and toward the solvent, and with the exception of Abu^16^ and Cys^20^, none of the α-helix residues are aligned to interact with the back surface of the β-sheet. A lack of H-bond donor and acceptor groups between the α-helix and β-sheet regions also makes the association between the α-helix and β-sheet strongly dependent on the presence of the disulfide bond between residues Cys^16^ and Cys^33^. This substitution also results in closer proximity of the Cys^2^–Cys^19^, Cys^5^–Cys^28^, and Cys^20^–Cys^35^ disulfide bonds, which may explain why multiple isomers occur with synthesis [29]. Ser-substitution of these residues, CTX3(S), however, preserves the overall structure. The elongation of the α-helix appears to stabilize interaction between the two Ser residues and an associated water molecule.

Substitution at the Cys^20^ and Cys^35^ residues appears to allow preservation of the overall structure with some degree of elongating of the α-helical region. Although the overall fold of the peptide is preserved, multiple H-bonds occur between the α-helix and β-sheets that do not occur in the native peptide, and the resulting fold places the Cys^2^–Cys^19^, Cys^5^–Cys^28^, and Cys^16^–Cys^33^ residues in close proximity and, like the Cys^16^–Cys^33^ substitution, may explain why multiple isomers occur with synthesis. The result for Ser-substitution is similar, except that the C-terminal antiparallel β-sheet is disrupted.

## 4. Materials and Methods

### 4.1. Initial Peptide Structures

All MD simulations were performed with the GROMACS 5.1.2 software packages using the CHARMM36m force field parameters with the CHARMM36m consistent version of TIP3P water [34,35,36,37,38,39,40,41,42,43,44,45,46]. The Lennard-Jones, electrostatic potentials, bond, bond angle, and torsional parameters for the Abu residue were assigned based on the similarity and transferability of force field parameters for similar noncyclic aliphatic residues (Ala, Leu, Ile, and Val) within the CHARMM36m force field using the same methodology as the CGenFF program and are provided in Appendix A [34,35,36,37,38]. The initial structure of CTX for the simulation was the first conformation of the published ^1^H NMR solution structure (Protein Data Bank ID: 1CHL) [30]. Starting structures of the CTX Abu- and Ser- substituted analogs as given in Table 1 were obtained by changing the respective residues from Cys to Abu or Ser using YASARA [47]. The protonation state and charges of all residues within the peptides were set to correspond to a pH of 7.0.

### 4.2. Molecular Dynamics

Peptides were solvated in dodecahedral boxes with TIP3P water with 150 mM NaCl. Additional Cl^–^ and Na^+^ ions were used to neutralize the charges of the systems. The minimum distance of the peptide to the edge of the dodecahedron was 1.4 nm, with the exception of CTX5(A) and CTX5(S), which required a 2.0 nm minimum distance of the peptide to the edge of the dodecahedron to prevent interaction of the peptide with its periodic image. The solvated systems were subjected to 5,000 steps of steepest descent energy minimization without restraints, allowing all bond distances and angles to relax. NVT (constant number, volume, and temperature) simulations of the positionally restrained peptides (force constant, 1000 kJ·mol^–1^) were performed for 10 nanoseconds at 310 K, followed by constant number, pressure and temperature (NPT) simulations of the positionally restrained peptides for 10 nanoseconds at 310 K and 101.325 kPa. The temperature and pressure, respectively, were kept constant by the stochastic velocity-rescaling method of Bussi et al. and the method of Berendsen et al. [48,49]. The relaxation constant was 0.1 picosecond and the isothermal compressibility was 4.5 × 10^−5^ bar^–1^. We used 2 femtoseconds for the integration step and the LINCS algorithm for constraining all bonds to their correct length; the warning angle was 30° [50,51]. The particle mesh Ewald method was used to calculate long-range electrostatic interactions; cutoff distances were 1.2 nm and 1.0 nm, and the Fourier spacing was 0.15 nm [52]. The switch method was used to calculate van der Waals interactions, the short-range and long-range cutoffs were 1.0 and 1.2 nm, respectively.

Production runs of 4.2 µs NPT simulations were performed at 310 K and 101.325 kPa. The peptides and solvent with ions were separately coupled to a Parrinello–Rahman barostat and the temperature of the peptides and solvent separately maintained by the stochastic velocity-rescaling method [48,53]. The integration step, bond and angle constraints, long-range electrostatic interactions, and van der Waals interactions were calculated as described above.

### 4.3. Trajectory Analysis

The first 200 nanoseconds of each simulation were considered system equilibration, and the subsequent 4 µs was used for analysis with a sampling frequency of 0.1 ns.

#### 4.3.1. Biophysical Properties

The Cα-trace RMSD from the ^1^H NMR solution conformation of CTX (RMSD_CTX_), Cα-trace RMSD from the average sampled peptide conformation (RMSD_AVG_), per-residue Cα-trace root mean square fluctuation (RMSF) from the average sampled peptide conformation, global and per-residue fraction of sampled secondary structure (α-helix, β-sheet, and β-bend/turn *g_rmsd, g_rmsf, and do_dssp,* utilities of GROMACS, respectively [31,45].

#### 4.3.2. Interactions

The intrachain sidechain–sidechain (SC–SC) and backbone–backbone (BB–BB) contacts and H-bonds were calculated with the *g_mdmat* and *g_hbond* utilities of GROMACS [44]. SC–SC and BB–BB contacts were designated for interatomic distances (between the heavy atoms of a residue) of 0.5 nm or less; H-bonds were designated for a donor-acceptor radius of 0.35 nm or smaller and a donor-hydrogen-acceptor angle of 30° or less. ^1^H NMR cannot measure the sulfur-sulfur interatomic distance, and assignment of the spectra and calculation of the peptide’s conformations is dependent on the presence of bonded CysHβ^1,2^–CysHβ^1,2^ nuclear Overhauser effect spectral peaks; therefore, interactions between bounded Cys, Abu-, or Ser-substituted residues were determined by measuring the CβHβ^1,2^–CβHβ^1,2^ center-of-mass distances and calculating the probability of contacts within the one-sided 95% confidence interval (CI_95_) [30,54]. The SASAs of residues 2, 5, 16, 19, 20, 28, 33, and 35 of CTX and its Abu- and Ser-substituted analogs where calculated using the *g_sasa* module of GROMACS using the atomic radii of Lee and Richards, a probe radius of 0.14 nm for water and 1000 points per sphere resolution [55].The residue SASAs were normalized to values of their maximal solvent-accessible surface area as determined from MD simulations on Acetyl-Ala-Xaa-Ala-NH_2_ (where Xaa is either Ser or Abu) and (Acetyl-Ala-Cys-Ala-NH_2_)_2_ (dimer with Cys–Cys disulfide bond) as outlined in Appendix A to yield relative SASA (rSASA) [56,57]. Normalization of SASA to the rSASA has been shown to correlate to models of protein folding, stability, and structural determination [57]. Hydration of the sidechains of Cys, Abu, and Ser was determined by integrating the radial distribution function for water molecules within 0.5 nm of the sidechain. The integral of this function is equal to the probability of finding water molecules within the defined radius [58,59].

#### 4.3.3. Conformational Analysis

The sampled conformations of the peptides were analyzed using the dihedral principal component analysis method [60,61]. The time-dependent φ/ψ dihedral angles from residues 2 to 35 of the peptides were extracted from the trajectories using the *g_rama* utility of GROMACS, and an in-house Python script was used to transform the data for the input to the dihedral principal component analysis program (provided by Dr. Yuguang Mu). We identified the lowest energy conformations by projecting the trajectories of the first 2 principal components onto a two-dimensional free energy (ΔG) landscape:(1)$$\Delta G=-R\cdot T\cdot ln\frac{\rho_{x,y}}{\rho_{max}}$$
where R is the universal gas constant, T is the temperature, and x and y are the first 2 dihedral principal components from the trajectory. The ΔG landscape was calculated by dividing the principal component 1 – principal component 2 subspace into grids to create a two-dimensional histogram of the sampled phase space and calculating the probability ρ_(x,y)_ using an in-house Python script, with ρ_max(x,y)_ corresponding to the grid with the maximum probability of occurrence. Results were visualized using the *scatterplot3D, akima*, and *latticeExtra* packages in the R software environment, with conformations and secondary structural elements rendered using YASARA [47,62,63,64,65]. Families of low-energy conformations were identified using k-means clustering as implemented in the *cluster* package in R, and the identified lowest energy conformations were extracted for further analysis [65,66]. The optimal number of clusters was determined by visual inspection, sum of squared error, average silhouette width, silhouette coefficient, and distribution plots [67,68,69,70].

#### 4.3.4. Essential Subspace Analysis

The overlap of the sampled essential subspace of trajectories for different peptides as a function of Cys–Cys substitution was determined by calculating the root-mean-square inner product (RMSIP_AB_):(2)$$RMSIP_{AB}={\left(\frac{1}{N}\cdot \overset{N}{\underset{i=1}{\sum }}\overset{N}{\underset{j=1}{\sum }}{\left(\eta_{i}^{A}\cdot \eta_{j}^{B}\right)}^{2}\right)}^{\frac{1}{2}}$$
comparing MD trajectories of peptides A and B, where $\eta_{i}^{A}$ and $\eta_{j}^{B}$ are the respective eigenvectors of the sampled essential subspace, and N is the total number of eigenvectors to be considered [71,72,73,74]. To correct for sampling errors and autocorrelation, the RMSIP can be normalized (nRMSIP):(3)$$nRMSIP=\frac{RMSIP_{A•B}}{\sqrt{RMSIP_{A1•A2}\cdot RMSIP_{B1•B2}}}$$

RMSIP_A1•A2_ and RMSIP_B1•B2_ are comparisons between the first and second halves of MD trajectories for peptides A and B, respectively. The nRMSIP ranges from 0 to 1; nRMSIP values approaching 1 indicate that the sampled essential subspaces of peptides A and B are similar, whereas lower values indicate differences between the sampled essential subspace of peptides A and B that cannot be explained by sampling alone. An nRMSIP of 0 indicates that the sampled essential subspaces of peptides A and B are orthogonal [74].

## 5. Conclusions

In CTX, the disulfide bonds between Cys^16^–Cys^33^ and Cys^20^–Cys^35^ residues are required to maintain the association between the α-helical region from residues 13 through 20 and the two antiparallel β-sheets from residues 27 through 29 and 32 through 34. This conformational dependence arises from the lack of SC–SC and BB–BB interactions between the three components of the αβ motif, either from hydrophobic, aromatic, ionic interactions, or H-bond formation. Of these two disulfide bonds, the bond between the Cys^16^ and Cys^33^ residues appears to be more critical, allowing for wider separation of the α-helix and β-sheet, whereas replacement of the Cys^20^ and Cys^35^ residues causes less conformational disruption. Selective use of Ser-substitution of residues 16 and 33 may mitigate this destabilizing effect, since an interaction may occur between the two residues and an associated water molecule. Although step-wise removal of individual disulfide bonds or all disulfide bonds increases protease degradation, it has a less marked effect on biological activity, as demonstrated by Ojeda et al. [29]. The binding region of CTX to its purported receptor may be independent of a restricted conformation. It should also be noted that even with removal of all four disulfide bonds, the peptides still share a substantial degree of conformational space sampling, and short, appropriately ordered segments of the peptide may present successfully to the receptor to initiate binding.

## Figures and Tables

**Figure 1 ijms-20-01261-f001:**
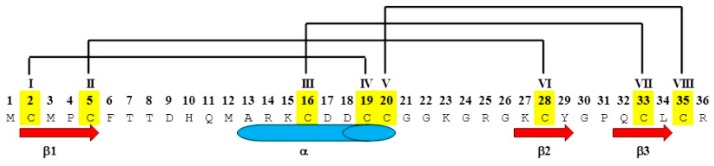
Amino acid sequence of Chlorotoxin (CTX), demonstrating the disulfide bond topology between Cys^2^–Cys^19^ (I–IV), Cys^5^–Cys^28^ (II–VI), Cys^16^–Cys^33^ (III–VII), and Cys^20^–Cys^35^ (V–VIII), shown as yellow with black brackets representing the associated disulfide bonds, and homology numbering shown as Roman numerals. The conserved secondary structural motifs of an α-helix from Ala^13^ to Cys^20^, a β-sheet from Cys^2^ to Cys^5^, and 2 antiparallel β-sheets from Lys^27^ to Tyr^29^ and Lys^32^ to Leu^34^ are shown as a blue cylinder and red arrows, respectively.

**Figure 2 ijms-20-01261-f002:**
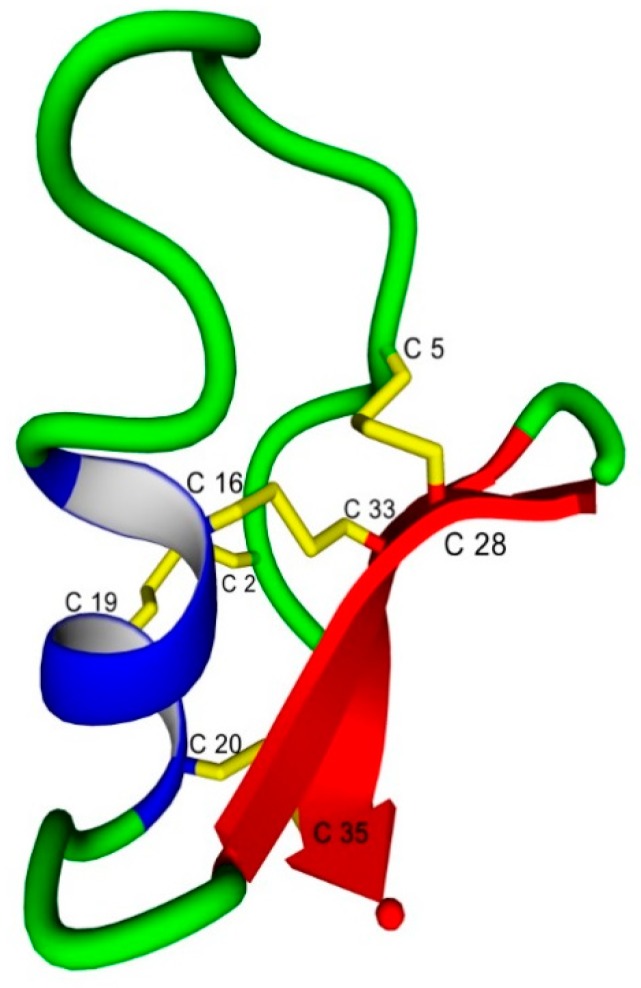
Solution conformation of chlorotoxin (CTX) as determined by ^1^H nuclear magnetic resonance spectroscopy (NMR) (Protein Data Bank ID: 1CHL) demonstrating the α-helix (Ala^13^ to Cys^20^) and 2 antiparallel β-sheets (Lys^27^ to Tyr^29^ and Lys^32^ to Leu^34^) with the four disulfide bonds between: Cys^2^–Cys^19^ (I–IV), Cys^5^–Cys^28^ (II–VI), Cys^16^–Cys^33^ (III–VII), and Cys^20^–Cys^35^ (V–VIII) residues, shown in yellow. The N-terminal β-sheet from Cys^2^ to Cys^5^ is not present in the lowest energy NMR conformation. Secondary structure elements are: α-helix, blue; β-sheet, red; and β-turn/bend/coil, green. Cys side chains and associated disulfide bonds, yellow.

**Figure 3 ijms-20-01261-f003:**
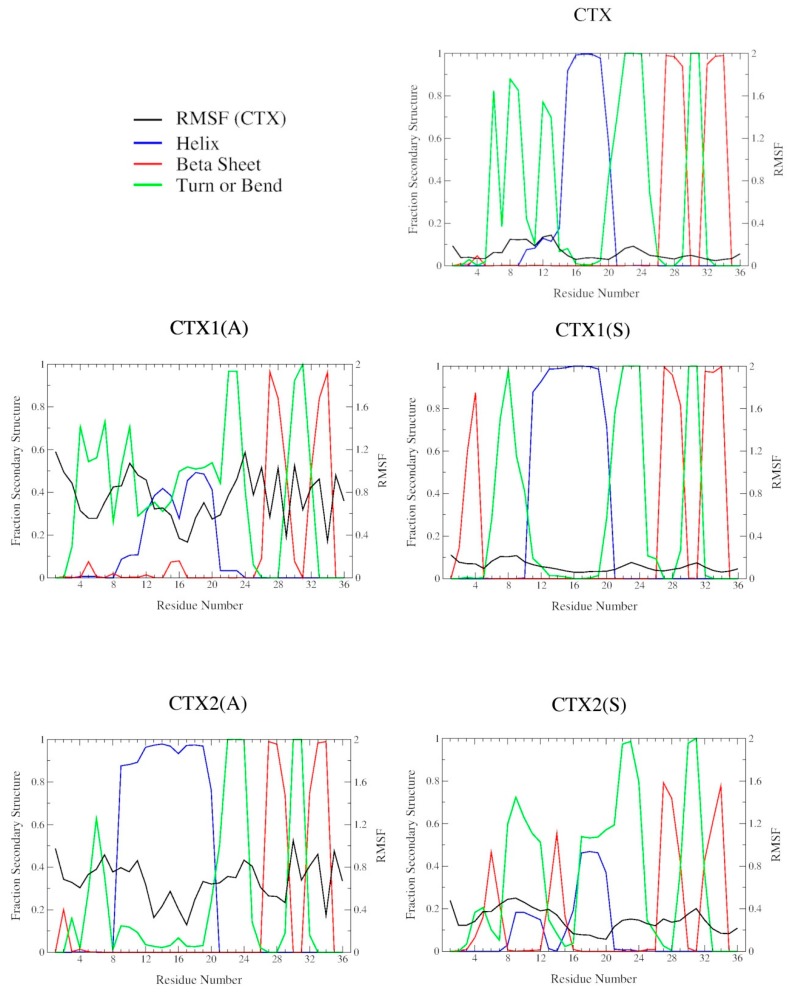
Fractions of sampled DSSP secondary structure (α-helix, blue; β-sheet, red; and β-turn/bend, green) as a function of residue number and Cα-trace root-mean-square fluctuation (RMSF) from the time average conformation of CTX (RMSF, black) for CTX and its Abu- and Ser-substituted analogs.

**Figure 4 ijms-20-01261-f004:**
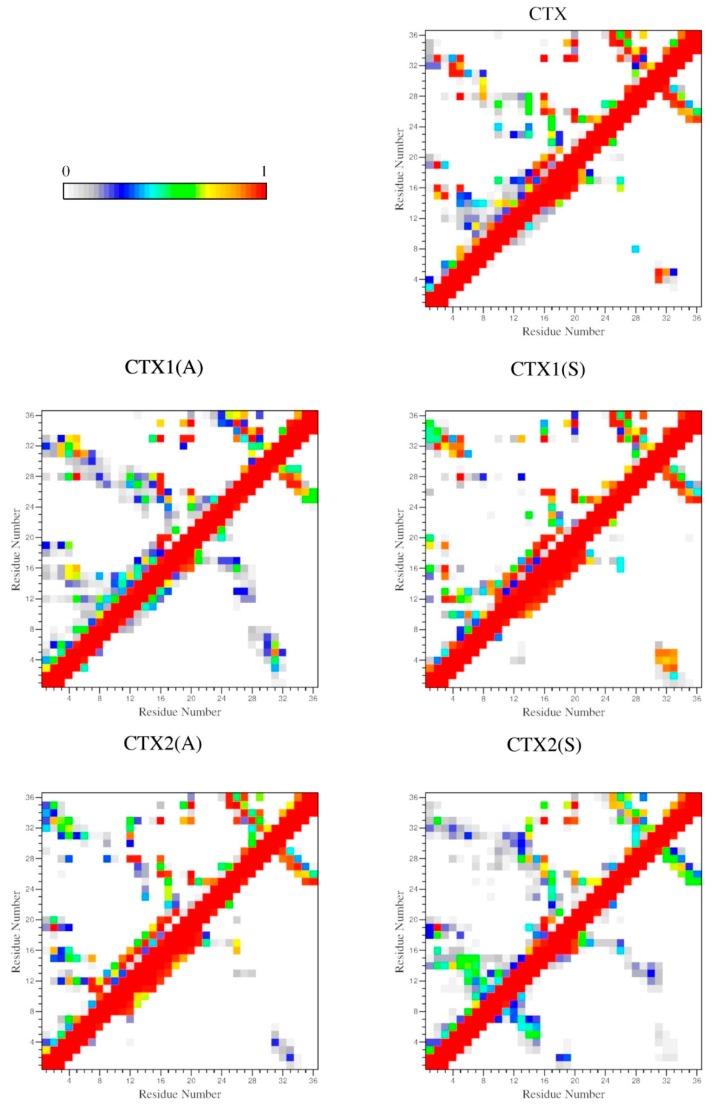
Probability maps for backbone–backbone contacts (below the diagonal) and sidechain–sidechain contacts (above the diagonal) as a function of residue number for CTX and its Abu- and Ser-substituted analogs. The color scale represents the relative probability of contact.

**Figure 5 ijms-20-01261-f005:**
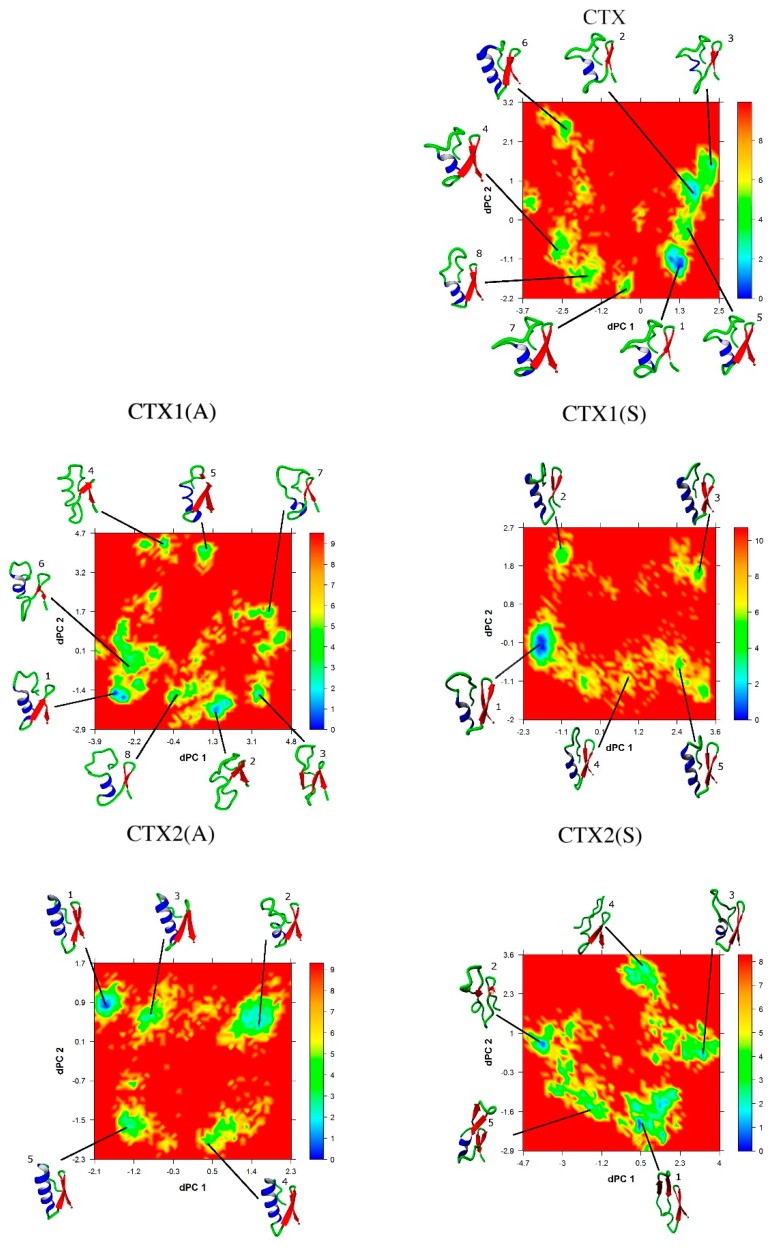
The free energy landscape (kJ·mol^–1^) as a function of the first two dihedral principal components (dPC1 and dPC2). The lowest energy conformations, as determined by cluster analysis, are shown and numbered in order from lowest to highest relative energy. The conformations have been rotated to optimally display the secondary structure elements and interactions. Secondary structure elements show as: red β-sheet, blue α-helix, and green turn/bend/coil.

**Figure 6 ijms-20-01261-f006:**
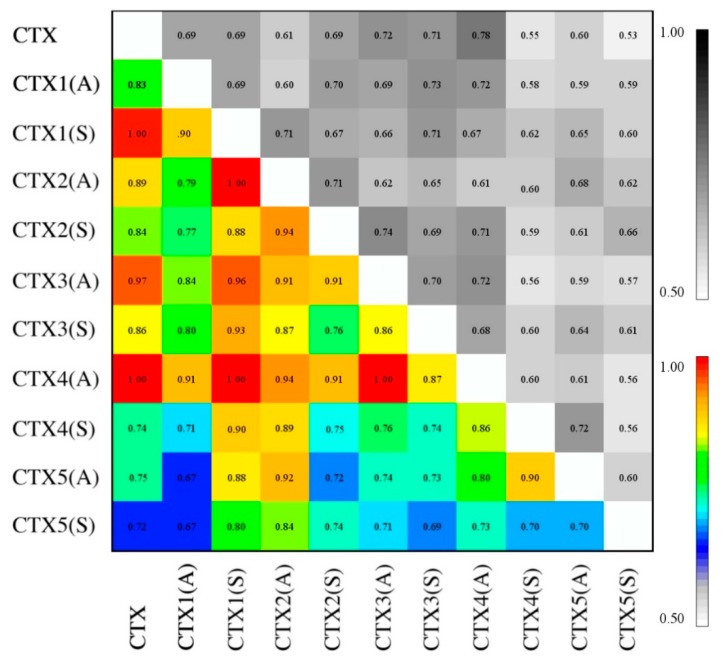
Comparison of essential subspace sampling. Root-mean-square inner product (RMSIP, top right in grayscale) and normalized RMSIP (bottom left in color) matrix of trajectories of molecular dynamics simulation of CTX and its Abu- and Ser- substituted analogs. By definition, the diagonal values equal 1.0, but not color-coded for clarity.

**Table 1 ijms-20-01261-t001:** Peptide naming convention for the Abu- and Ser-substituted chlorotoxin analogs. Native CTX sequence and disulfide bond homology numbering scheme are shown in Figure 1 and Figure 2.

Peptide	Abu Substitution	Ser-Substitution	Homology Disulfide Bond(s) Removed
CTX			None
CTX1(A)	2, 19		I–IV
CTX1(S)		2, 19	I–IV
CTX2(A)	5, 28		II–VI
CTX2(S)		5, 28	II–VI
CTX3(A)	16, 33		III–VII
CTX3(S)		16, 33	III–VII
CTX4(A)	20, 35		V–VIII
CTX4(S)		20, 35	V–VII
CTX5(A)	2, 5, 16, 19, 20, 28, 33, 35		All
CTX5(S)		2, 5, 16, 19, 20, 28, 33, 35	All

**Table 2 ijms-20-01261-t002:** Calculated (mean ± SD) Cα-trace RMSDs between the peptide and the average conformation of CTX, between the Abu- or Ser-substituted analog and the average conformation of itself, and the fraction (ρ) of sampled DSSP secondary structure.

Peptide	RMSD/nm		ρ		
	CTX ^a^	AVG ^b^	α-Helix	β-Sheet	β-Turn/Bend
CTX	0.73 ± 0.02	0.73 ± 0.02	0.31 ± 0.05	0.16 ± 0.02	0.17 ± 0.05
CTX1(A)	0.71 ± 0.02	0.74 ± 0.02	0.39 ± 0.09	0.14 ± 0.05	0.11 ± 0.09
CTX1(S)	0.73 ± 0.02	0.73 ± 0.02	0.27 ± 0.04	0.20 ± 0.03	0.26 ± 0.02
CTX2(A)	0.73 ± 0.02	0.73 ± 0.02	0.23 ± 0.07	0.16 ± 0.03	0.31 ± 0.04
CTX2(S)	0.73 ± 0.02	0.74 ± 0.02	0.34 ± 0.08	0.16 ± 0.08	0.08 ± 0.07
CTX3(A)	0.73 ± 0.02	0.76 ± 0.02	0.31 ± 0.07	0.19 ± 0.06	0.13 ± 0.14
CTX3(S)	0.73 ± 0.02	0.72 ± 0.02	0.29 ± 0.06	0.13 ± 0.06	0.22 ± 0.08
CTX4(A)	0.73 ± 0.02	0.72 ± 0.02	0.32 ± 0.08	0.16 ± 0.04	0.19 ± 0.07
CTX4(S)	0.72 ± 0.02	0.72 ± 0.02	0.30 ± 0.06	0.14 ± 0.04	0.27 ± 0.06
CTX5(A)	0.72 ± 0.02	0.61 ± 0.02	0.28 ± 0.09	0.03 ± 0.05	0.21 ± 0.14
CTX5(S)	0.73 ± 0.02	0.65 ± 0.02	0.30 ± 0.09	0.05 ± 0.06	0.09 ± 0.10

^a^ Cα-trace comparison between the peptide and the average (AVG) conformation of CTX; ^b^ Cα-trace comparison between the peptide and the average (AVG) conformation of itself.

**Table 3 ijms-20-01261-t003:** The CβHβ^1,2^–CβHβ^1,2^ center-of-mass distances (D) in nm (mean±SD) between residue pairs for CTX and its Abu- and Ser-substituted analogs given in nm. Probabilities (ρ) of contact ≤ 0.26 nm are in parenthesis. The Abu- and Ser-substituted residues are highlighted with grey background ^a^

	D (ρ)			
Peptide	2–19	5–28	16–33	20–35
CTX	0.24 ± 0.01 (0.98)	0.21 ± 0.01 (0.99)	0.26 ± 0.01 (0.71)	0.21 ± 0.01 (1.00)
CTX1(A)	0.45 ± 0.08 (0.02)	0.22 ± 0.02 (1.00)	0.21 ± 0.01 (1.00)	0.21 ± 0.01 (1.00)
CTX1(S)	0.43 ± 0.04 (0.00)	0.22 ± 0.02 (1.00)	0.21 ± 0.01 (1.00)	0.21 ± 0.01 (1.00)
CTX2(A)	0.22 ± 0.02 (0.97)	0.49 ± 0.08 (0.00)	0.24 ± 0.02 (0.84)	0.21 ± 0.01 (1.00)
CTX2(S)	0.23 ± 0.02 (0.89)	0.46 ± 0.11 (0.01)	0.21 ± 0.02 (0.98)	0.21 ± 0.01 (1.00)
CTX3(A)	0.24 ± 0.02 (0.67)	0.22 ± 0.01 (1.00)	0.33 ± 0.07 (0.13)	0.21 ± 0.01 (1.00)
CTX3(S)	0.24 ± 0.02 (0.77)	0.22 ± 0.01 (0.99)	0.26 ± 0.05 (0.70)	0.21 ± 0.01 (1.00)
CTX4(A)	0.22 ± 0.02 (0.99)	0.22 ± 0.01 (1.00)	0.24 ± 0.02 (0.80)	0.22 ± 0.03 (0.90)
CTX4(S)	0.23 ± 0.01 (0.99)	0.21 ± 0.01 (1.00)	0.23 ± 0.01 (1.00)	0.30 ± 0.08 (0.39)
CTX5(A)	0.62 ± 0.28 (0.04)	0.65 ± 0.22 (0.01)	0.55 ± 0.18 (0.01)	0.62 ± 0.19 (0.02)
CTX5(S)	0.78 ± 0.30 (0.01)	0.69 ± 0.28 (0.05)	0.70 ± 0.21 (0.00)	0.75 ± 0.25 (0.00)

^a^ The CβHβ^1,2^–CβHβ^1,2^ center-of-mass distance for the (Acetyl-Ala-Cys-Ala-NH_2_)_2_ dipeptide, as described in Appendix A; D was 0.23 ± 0.03 nm and one-sided CI_95_ was ≤ 0.26 nm.

**Table 4 ijms-20-01261-t004:** Probability of finding water molecules within 0.5 nm of the terminal sidechain γ atoms of Cys-, Abu-, and Ser- substituted residues (Sγ, CγHγ, and OγHγ, respectively). The relative solvent-accessible surface area (rSASA) of the residues are in parenthesis. The Abu- and Ser-substituted residues are highlighted with grey background ^a,b,c,d,e^

Peptide	Residue							
	2	5	16	19	20	28	33	35
CTX	0.066	0.044	0.013	0.076	0.020	0.049	0.019	0.094
	(0.27)	(0.09)	(0.02)	(0.38)	(0.14)	(0.10)	(0.11)	(0.45)
CTX1(A)	0.173	0.016	0.012	0.113	0.022	0.037	0.017	0.086
	(0.79)	(0.08)	(0.01)	(0.50)	(0.15)	(0.11)	(0.15)	(0.44)
CTX1(S)	0.147	0.070	0.028	0.121	0.018	0.087	0.041	0.081
	(0.56)	(0.16)	(0.01)	(0.55)	(0.15)	(0.41)	(0.04)	(0.38)
CTX2(A)	0.077	0.144	0.013	0.084	0.015	0.060	0.012	0.081
	(0.24)	(0.56)	(0.01)	(0.42)	(0.16)	(0.30)	(0.07)	(0.41)
CTX2(S)	0.086	0.186	0.033	0.066	0.027	0.104	0.032	0.079
	(0.27)	(0.50)	(0.04)	(0.27)	(0.17)	(0.30)	(0.17)	(0.40)
CTX3(A)	0.090	0.035	0.078	0.059	0.025	0.044	0.053	0.070
	(0.33)	(0.10)	(0.15)	(0.23)	(0.15)	(0.18)	(0.24)	(0.32)
CTX3(S)	0.079	0.032	0.050	0.061	0.029	0.048	0.060	0.086
	(0.37)	(0.09)	(0.02)	(0.30)	(0.17)	(0.22)	(0.14)	(0.41)
CTX4(A)	0.075	0.032	0.031	0.093	0.038	0.052	0.024	0.111
	(0.22)	(0.06)	(0.02)	(0.46)	(0.15)	(0.25)	(0.07)	(0.49)
CTX4(S)	0.082	0.006	0.024	0.076	0.046	0.026	0.025	0.165
	(0.33)	(0.05)	(0.00)	(0.48)	(0.13)	(0.20)	(0.03)	(0.56)
CTX5(A)	0.101	0.138	0.101	0.138	0.126	0.124	0.123	0.138
	(0.69)	(0.61)	(0.35)	(0.64)	(0.58)	(0.54)	(0.53)	(0.59)
CTX5(S)	0.203	0.194	0.156	0.166	0.202	0.172	0.182	0.204
	(0.76)	(0.72)	(0.44)	(0.65)	(0.75)	(0.53)	(0.62)	(0.71)

^a^ The probability of finding water molecules within 0.5 nm of the terminal γ-sidechain atom of Cys (Sγ) in the (Acetyl-Ala-Cys-Ala-NH_2_)_2_ dipeptide, as described in Appendix A Materials, is 0.114; ^b^ The probability of finding water molecules within 0.5 nm of the terminal γ-sidechain atoms of Abu (CγHγ) in the Acetyl-Ala-Abu-Ala-NH_2_ peptide, as described in Appendix A, is 0.190; ^c^ The probability of finding water molecules within 0.5 nm of the terminal γ-sidechain atoms of Ser (OγHγ) in the Acetyl-Ala-Ser-Ala-NH_2_ peptide, as described in Appendix A, is 0.213; ^d^ The rSASA is the ratio of the residue solvent exposed surface area (SASA) to the residue maximum solvent exposed surface area (mSASA) in (Acetyl-Ala-Xaa-Ala-NH_2_) peptide where Xaa is either Abu or Ser as described in the Methods and Appendix A. The mSASA for Abu was: 1.347 ± 0.121 nm^2^. The mSASA for Ser was: 1.219 ± 0.111 nm^2^; ^e^ The rSASA for Cys is the ratio of the residue solvent exposed surface area (SASA) to the residue maximum solvent exposed surface area (mSASA) in (Acetyl-Ala-Cys-Ala-NH_2_)_2_ dipeptide as described in the Methods and Appendix A. The mSASA for Cys was: 0.758 ± 0.142 nm^2^.

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
