# Peer review of "Effects of Selective Substitution of Cysteine Residues on the Conformational Properties of Chlorotoxin Explored by Molecular Dynamics Simulations"

_ijms, 2019, doi:10.3390/ijms20061261_

Round 1

Reviewer 1 Report

The authors have prepared a completely revised, new version of the former manuscript addressing very satisfactorily all the concerns raised in my previous review. The new version is a significant improvement of the former manuscript. In its current form, it is a readable report that now warrants publication in the IJMS.

Only a very minor issue. Please correct the name of the LJ potential: it is “Lennard-Jones”, no “Leonard-Jones” as it incorrectly reads in lines 328 and 454.

Reviewer 2 Report

I have no further comments about the revised manuscript.

This manuscript is a resubmission of an earlier submission. The following is a list of the peer review reports and author responses from that submission.

Round 1

Reviewer 1 Report

Gregory et al. have performed molecular dynamics (MD) simulations of the wild-type chlorotoxin (CTX) and its mutants (listed in Table 1) to investigate the role of the disulfide bonds in stabilizing the CTX structure. Emphasis has been placed on investigating the role of hydrophobic (Abu) versus hydrophilic (Ser) isosteric substitutions, the distribution of the hydration shell around each respective residue substitution, and subsequent changes in the secondary and tertiary conformations.

    The research is designed appropriately. The MD simulations seem to be performed appropriately. The analyses of the MD trajectories are adequate for the problem under study. The conclusions are supported by the results.

    The manuscript reads well and may be of potential interest to readers of the International Journal of Molecular Sciences.

Reviewer 2 Report

This manuscript (MS) addresses an interesting problem: investigate in silico the effect of selective changes of cysteines on the structural and stability properties of chlorotoxin (CTX). Given the pharmacological interest of this peptide, its structural study is all the more relevant. Besides, the methodology employed, in depth analyses of molecular dynamics (MD) trajectories, is appropriate. And in fact, the Conclusions section 5 presents some valuable remarks.

The big problem with the current form of the MS is that it is completely unreadable. It is an exhaustive and exhausting collection of data, very small figures, tons of numbers… all without any effort of being analyzable. It is virtually impossible to follow the presentation of results.

It would be much more useful (and easier!) to present relevant differences between peptides besides focusing on the really important properties. In this regard, features as SASA or Rg are not as relevant as RMSF, for example.

The notation used throughout the whole MS is clumsy and it does not help the reader to read it. “Abu” (or “Ser”) and then 8 numbers in superscript does not seem a reasonable notation for a peptide when this notation is repeated many times along the text. Representing a peptide with other symbol instead of e.g. [Abu^2^19]CTX and using one-letter instead of three-letter symbols for amino acids would also help.

Other issues that the authors should address in their revised version:

* The method used to assign secondary structure is not indicated (it seems it is DSSP but this is not explicitly stated).

* The reliability of the parameterization of Abu should be assessed in some way for example, presenting some calibration calculations. But Table A1 is irrelevant.

* The method used to calculate configurational entropies (Fig. A1) is not indicated. Thermodynamic relationships underlying the usual methods to obtain entropy from PCA of MD trajectories are valid only in the NVT ensemble. Since the only reference that apparently has some relation with this issue is Ref. 69, it seems that the authors have employed the Andricioaei-Karplus method instead of the common Schlitter method. This must be clarified.

Summarizing, it is very likely that this MS could be of interest to the IJMS readership but not obviously in its current form. Only after a thorough revision, a complete reorganization of all the material, and a drastic shortening (the current PDF has 36 pages!) I could properly assess whether it really is.

Reviewer 3 Report

Well described molecular modeling work

Caption 3:  average